# On the Capacitated Facility Location Problem with Scarce Resources

**Gennaro Auricchio**[1]          **Harry J. Clough**[1]          **Jie Zhang**[1]

[1]Computer Science Department, University of Bath, Bath, Somerset, England, UK

## Abstract

This paper investigates the Mechanism Design aspects of the $m$-Capacitated Facility Location Problem where the total facility capacity is lower than the number of agents. Following Aziz et al. [2020b], the Social Welfare of the facility location is determined through a First-Come-First-Served (FCFS) game where agents compete after the facility positions are established. When the number of facilities is $m > 1$, the Nash Equilibrium (NE) of the FCFS game is not unique, thus the utility of the agents and the notion of truthfulness are not well-defined. To address these issues, we consider absolutely truthful mechanisms, i.e. mechanisms able to prevent agents from misreporting regardless of the strategies played during the FCFS game. We pair this more stringent truthfulness requirement with the notion of Equilibrium Stable (ES) mechanism, i.e. mechanisms whose Social Welfare does not depend on the NE of the FCFS game. We show that the class of percentile mechanisms is absolutely truthful and characterize under which conditions they are ES. We then show that the approximation ratio of each ES percentile mechanism is bounded and determine its value. Notably, when all the facilities have the same capacity and the number of agents is large enough, it is possible to achieve an approximation ratio smaller than $1 + \frac{1}{2m-1}$. We enhance our findings by empirically evaluating the mechanisms' performances when agents' true positions follows a distribution.

## 1 INTRODUCTION

The $m$-Capacitated Facility Location Problem ($m$-CFLP) is a generalization of the $m$-Facility Location Problem ($m$-FLP) in which each facility has a maximum service ca-

pacity (Brimberg et al. [2001], Pal et al. [2001], Aardal et al. [2015]). Both the $m$-FLP and the $m$-CFLP are crucial sub-problems in social choice theory, such as disaster relief (Balcik and Beamon [2008]), supply chain management (Melo et al. [2009]), healthcare (Ahmadi-Javid et al. [2017]), clustering (Hastie et al. [2009], Auricchio et al. [2019]), and public facility accessibility (Barda et al. [1990]). In its fundamental guise, the $m$-CFLP consists of locating $m$ facilities amidst $n$ self-interested agents. Each facility has a capacity limit, which describes the maximum amount of agents it can serve. Since every agent requires access to the facility, they prefer to have a facility located as close as possible to their position. In this scenario, optimizing a communal goal solely based on reported preferences leads to undesirable manipulation driven by the agents' self-interested behaviour. For this reason, a key property that a mechanism should possess is *truthfulness*, which ensures that no agent can gain an advantage by misrepresenting their private information. This stringent property, however, forces the mechanism to produce suboptimal locations, leading to an efficiency loss which is quantified by the *approximation ratio* – that is the worst-case ratio between the objective achieved by the mechanism and the optimal objective attainable (Nisan and Ronen [1999]). Defining efficient routines that forbid agents from manipulating is a key problem in mechanism design.

In this paper, we study the mechanism design aspects of the $m$-CFLP. In particular, we focus on the framework presented in (Aziz et al. [2020b]), where we have $m$ facilities whose total capacity is lower than the number of agents needing accommodation. Moreover, the mechanism designer cannot force agents to use a specific facility, therefore the agents compete in a First-Come-First-Served (FCFS) game to determine who is accommodated by the facilities. The overall process therefore consists of two parts. First, the agents report their position to a mechanism, which locates the facilities. Second, the agents compete in the FCFS game to determine their utilities. Notice that the Social Welfare achieved by the mechanism and the utilities of the agents depend on the Nash Equilibria (NE) of the FCFS game in-

duced by the facilities' placements. When we need to place a single facility, that is $m = 1$, the FCFS game has always a unique NE. When $m \geq 2$, however, the NE of the FCFS game is no longer unique, posing a series of challenges: if the NE is not unique, the agents' utilities and thus the Social Welfare achieved by a facility placement have different values depending on the equilibrium of the FCFS game. As a consequence, the approximation ratio also depends on the specific NE. Furthermore, the classic definition of the truthful mechanism is no longer suitable for this problem as it does not consider the different strategies that the agents may adopt during the FCFS game. Addressing these issues are a major challenge for this problem, and thus they were left as an open problem in (Aziz et al. [2020b]).

**Our Contribution.** In this paper, we study the mechanism design aspects of the $m$-CFLP when the total capacity of the facilities is less than the number of agents. In particular, we extend the framework presented in (Aziz et al. [2020b]) to encompass problems in which there is more than one capacitated facility to locate. First, we show that, regardless of how we locate the facilities, the FCFS game induced by the location has at least one pure NE. We then present a notion of truthfulness that accounts for the different strategies the agents can adopt during the FCFS game, which we name *absolute truthfulness*. Finally, we introduce the class of Equilibrium Stable (ES) mechanisms, i.e. mechanisms whose output induces a FCFS game in which every NE achieves the same Social Welfare.

Within this framework, we study the percentile mechanisms (Sui et al. [2013]). We show that every percentile mechanism is absolutely truthful. We then characterize the set of conditions under which the a percentile mechanism is ES and compute their approximation ratio.

First, we consider the case $m = 2$ and show that an absolutely truthful and ES percentile mechanism exists. We then characterize the approximation ratio of a percentile mechanism as a function of the facilities' capacity and to the vector inducing the mechanism. As a consequence, we determine the best percentile mechanism as a function of the number of agents $n$ and the capacities of the facilities $k_1$ and $k_2$. In particular, we show that the best approximation ratio that an ES percentile mechanism placing two facilities can achieve is $\frac{4}{3}$, which occurs when $k_1 = k_2 = k$ and $n \geq 3k$.

Second, we consider the case in which $m > 2$. In this framework, a percentile mechanism that is ES and places the facilities at more than two different locations might not exist. However, when all the facilities have the same capacity and $n \geq (2k - 1)m$ holds, there exists a percentile mechanism whose approximation ratio is less than $1 + \frac{1}{m-1}$. This result has two interesting consequences: (i) it shows that, under suitable assumptions, the percentile mechanisms are asymptotically optimal with respect to $m$ and (ii) it highlights the

differences between the $m$-CFLP and the $m$-FLP. Indeed, in the classic framework, any percentile mechanism has unbounded approximation ratio whenever $m > 2$, (Walsh [2020], Fotakis and Tzamos [2014]). Due to space limits, some of the proofs are deferred to the Appendix.

Lastly, we empirically study the behaviour of the best percentile mechanisms under the assumption that the agents are distributed according to a distribution $\mu$. We focus on the case in which we have two facilities, since it is the case in which the gap between 1 and the approximation ratio of the best possible percentile mechanism is largest. From our analysis, we observe that when the agents follow a distribution, the performances of the ES mechanism are close to optimal, regardless of the distribution.

**Related Work.** The Mechanism Design aspects of the $m$-FLP were firstly studied in (Procaccia and Tennenholtz [2013]). Following this seminal work, various mechanisms with constant approximation ratios for placing one or two facilities on lines Filos-Ratsikas et al. [2017], trees (Feldman and Wilf [2013], Filimonov and Meir [2021]), circles (Lu et al. [2010, 2009]), general graphs Alon et al. [2010], Dokow et al. [2012], and metric spaces Meir [2019], Tang et al. [2020] were introduced. Crucially, all these positive results are limited to scenarios where the number of agents is restricted or the number of facilities is either 1 or 2. For a comprehensive survey of the mechanism design aspects of the FLP, we refer to (Chan et al. [2021]).

The $m$-Capacitated Facility Location Problem ($m$-CFLP) is a variation of the $m$-FLP in which each facility has a capacity limit. The first game theoretical framework for the $m$-CFLP was presented in (Aziz et al. [2020a]). In this paper, the authors studied the case in which there are at least two facilities whose total capacity is enough to accommodate all the agents and the mechanism designer has to decide where to place the facilities and which agent can use them. Following this initial study, in (Walsh [2022]) the authors proposed a more theoretical analysis of the problem, while in (Auricchio et al. [2024a]) it was shown that it is possible to define deterministic mechanisms with bounded approximation ratio when all the facilities have the same capacities and the number of agents is equal to the total capacities of the facilities. Lastly, in (Auricchio et al. [2024b]) the $m$-CFLP is studied from a Bayesian mechanism design perspective.

In this paper, we consider an alternative game theoretical framework for the $m$-CFLP, firstly introduced in (Aziz et al. [2020b]). This framework differs from the one proposed in (Aziz et al. [2020a]) for two main reasons: (i) the total capacity of the facilities is lower than the total number of agents, thus part of the agents cannot be accommodated and (ii) the mechanism designer does not enforce an agent-to-facility assignment. Thus, after the positions of the facilities are elicited, the agents compete in a First-Come-First-Served

(FCFS) game to access the facilities. When $m = 1$, the FCFS game is trivial as the agents accommodated by the facility are the ones that are closer to the facility. When $m > 1$, designing mechanisms becomes much more complicated as, for example, the Nash Equilibrium (NE) of the FCFS game is no longer unique. As a consequence the utility of every agent depends on the specific NE of the game, making the classic notion of trustfulness unfit for this framework.

## 2 SETTING STATEMENT

Let $\vec{x} \in [0, 1]^n$ be the position of $n$ agents in the interval $[0, 1]$. We denote with $\vec{k} = (k_1, \ldots, k_m)$ the $m$-dimensional vector containing all the capacities of the $m$ facilities, so that $k_j$ is the maximum number of agents that the $j$-th facility can accommodate. We assume that the total capacity of the facilities is less than the number of agents, hence $\sum_{j \in [m]} k_j < n$. In this case, a mechanism is a function $M : [0, 1]^n \to \mathbb{R}^m$ that maps a vector containing the agents' reports to a facility location $\vec{y} = (y_1, \ldots, y_m)$, where $y_j$ is the position of the facility with capacity $k_j$. After the mechanism places the facilities, agents compete in a First-Come-First-Served (FCFS) game to access the facilities.

**First-Come-First-Served Game.** Let $\vec{y} = (y_1, \ldots, y_m)$ be a vector containing the position of the facilities to locate, that is the facility with capacity $k_j$ is located at $y_j$. Then, given $\vec{x} \in [0, 1]^n$ the vector containing the positions of the $n$ agents on the interval $[0, 1]$, the FCFS game induced by the facility location $\vec{y}$ is as follows:

(i) Each agent selects one of the facilities. Thus the set of strategies of each agent is the set $[m] := \{1, 2, \ldots, m\}$. We denote with $\vec{s} \in [m]^n$ the vector containing a set of pure strategies. For every $\vec{s}$, we denote with $\mathcal{S}_j \subset [n]$ the set of agents that selected strategy $j$.

(ii) Denoted with $d_{i,j} = |x_i - y_j|$ the distance of the agent $i$ from the location of the facility they selected, we define $T_j \subset \mathcal{S}_j$ as the set containing the agents in $\mathcal{S}_j$ whose value $d_{i,j}$ is in among the $k_j$ lowest values. Break ties according to the prefixed priority rule.

(iii) Finally, the utility of agent $i$ is defined as follows

$$u_i(\vec{x}, \vec{y}; \vec{s}) = \begin{cases} 1 - |x_i - y_j| & \text{if} \quad i \in T_j \\ 0 & \text{otherwise} \end{cases}.$$

First, we show that every FCFS game has at least one pure Nash Equilibrium (NE).

**Theorem 1.** For every $\vec{x} \in [0, 1]^n$, every $\vec{y} \in [0, 1]^m$, and every capacity vector $\vec{k}$, the related FCFS game admits at least one pure Nash Equilibrium.

When the vector $\vec{k}$ is clear from the context, we denote the set of all the pure Nash Equilibria with $NE(\vec{x}, \vec{y})$. The

Social Welfare (SW) of the facility location $\vec{y}$ according to $\gamma \in NE(\vec{x}, \vec{y})$ is defined as the sum of all the agents' utilities, that is $SW_\gamma(\vec{x}, \vec{y}) = \sum_{i \in [n]} u_i(\vec{x}, \vec{y}; \gamma)$. Notice that when $m = 1$, the Nash Equilibrium of the FCFS game is unique, since every agent can play only one strategy, hence the SW of the game is well defined. This is no longer true when we need to place more than one facility. Moreover, when $m > 1$, the SW of the game changes depending on the specific Nash Equilibrium.

**Example 1.** Let us consider the case in which we have 5 agents and need to place two facilities with $k_1 = k_2 = 2$. Let $\vec{x} := (0, 0.3, 0.4, 0.5, 0.9) \in [0, 1]^5$ be the vector containing the agents' positions. If $\vec{y} = (0.3, 0.5)$, both $\gamma_1 = (1, 1, 2, 2, 2)$ and $\gamma_2 = (1, 1, 1, 2, 2)$ are pure NE of the FCFS game. However, the SW of the FCFS game depends on the specific NE, indeed $SW_{\gamma_1}(\vec{x}, \vec{y}) = 3.6 > 3.5 = SW_{\gamma_2}(\vec{x}, \vec{y})$. Moreover, the utility the agent located at 0.9 is zero or 0.6 depending on the equilibrium.

We define the optimal SW achievable on the instance $\vec{x}$ as

$$SW_{opt}(\vec{x}) = \sup_{\vec{y} \in [0,1]^m} \sup_{\gamma \in NE(\vec{x}, \vec{y})} SW_\gamma(\vec{x}, \vec{y}). \quad (1)$$

**Mechanism Design Aspects.** A key property that a mechanism $M$ must possess is truthfulness, which ensures that no agent can increase its utility by misreporting their position. As shown in Example 1, the utility of an agent depends on the other agents' strategies. For this reason, we employ a stronger notion of truthfulness that keeps track of the different strategies that agents can play in the FCFS game.

**Definition 2.1.** A mechanism $M$ is absolutely truthful if no agent increases its utility by misreporting, regardless of the strategies played by other agents. More formally, for every $i \in [n], \vec{x} \in [0, 1]^n$, and $\vec{s}_{-i} \in [m]^{n-1}$, we have

$$\max_{s_i \in [m]} u_i(\vec{x}, M(\vec{x}); s_i, \vec{s}_{-i}) \geq \max_{s_i' \in [m]} u_i(\vec{x}, M(\vec{x}'); s_i', \vec{s}_{-i}),$$

where (i) $\vec{x}' = (x_i', \vec{x}_{-i})$ for every $x_i' \in [0, 1]$, (ii) $\vec{x}_{-i}$ and $\vec{s}_{-i}$, are the vectors containing the positions and strategies of the other $n - 1$ agents, respectively.

As for the optimal solution, the FCFS game induced by the output of a mechanism has multiple NE, hence the SW of the mechanism is not unique. For this reason, we introduce the notion of Equilibrium Stable mechanism.

**Definition 2.2.** An absolutely truthful mechanism $M$ is said Equilibrium Stable (ES) if, for every $\vec{x}$, we have that

$$SW(\vec{x}, M(\vec{x}); \gamma) = SW(\vec{x}, M(\vec{x}); \gamma')$$

for every Nash Equilibria $\gamma, \gamma' \in NE(\vec{x}, M(\vec{x}))$.

An absolutely truthful mechanism is not necessarily ES. For example, consider the mechanism that places the facilities

at $(0.3, 0.5)$ regardless of the agents' reports. No agent can manipulate the outcome of the mechanism by changing their reports, however, going back to Example 1, we have that the mechanism induces two NE with different SW on the instance $(0, 0.3, 0.4, 0.5, 0.9)$. Finally, since we are considering the utilities of the agents, we define the approximation ratio of an ES mechanism as the worst-case ratio of the optimal SW and the SW achieved by the mechanism.

**Definition 2.3.** Let $M$ be an absolutely truthful and ES mechanism. We define the approximation ratio of $M$ as

$$ar(M) = \sup_{\vec{x} \in [0,1]^n} \frac{SW_{opt}(\vec{x})}{SW_M(\vec{x})},$$

where $SW_M$ is the SW value achieved by $M$ on any of the NE in $NE(\vec{x}, M(\vec{x}))$ and $SW_{opt}$ is defined in (1).

# 3 ABSOLUTELY TRUTHFUL AND ES MECHANISMS TO PLACE MORE THAN ONE FACILITY

In this section, we study the class of percentile mechanisms and characterize under which conditions a percentile mechanism is absolutely truthful and ES.

**Definition 3.1** (Percentile Mechanism, (Sui et al. [2013])). Given a percentile vector $\vec{v} \in [0, 1]^m$, the routine of the percentile vector associated with $\vec{v}$, namely $\mathcal{PM}_{\vec{v}}$, is as follows: (i) The mechanism designer collects all the reports of the agents $\{x_1, x_2, \ldots, x_n\}$ and sorts them in non-decreasing order, so that $x_i \leq x_{i+1}$. (ii) The mechanism then places the $m$ facilities at the positions $y_j = x_{i_j}$, where $i_j = \lfloor (n-1)v_j \rfloor + 1$ for every $j \in [m]$.

All percentile mechanisms are absolutely truthful.

**Theorem 2.** $\mathcal{PM}_{\vec{v}}$ is absolutely truthful for every $\vec{v}$.

*Proof.* Toward a contradiction, let $\mathcal{PM}_{\vec{v}}$ be a percentile vector such that, on instance $\vec{x} \in [0, 1]^m$, the agent whose real position is $x_i$ can manipulate by reporting $x'_i \in [0, 1]$. Let $\vec{y}$ be the output of $\mathcal{PM}_{\vec{v}}$ in the truthful input and let $\vec{y}'$ be the output of $\mathcal{PM}_{\vec{v}}$ after the agent manipulation. If $x'_i \leq x_i$, we have that the position of the facilities that $\mathcal{PM}_{\vec{v}}$ places on the left of $x_i$ move further to the left of $x_i$. Each facility that was placed by $\mathcal{PM}_{\vec{v}}$ on the right of $x_i$ does not change position. Thus it holds $|x_i - y_j| \leq |x_i - y'_j|$ for every $j \in [m]$. Finally, let $\vec{s}_{-i} \in [m]^{n-1}$ be a vector containing the strategies of the other agents and let us define with $F_i(\vec{z}) \subset [m]$ the set of strategies that give a non-null utility to the agent at $x_i$ when the facilities are located at $\vec{z}$. Since $|x_i - y_j| \leq |x_i - y'_j|$ for every $j \in [m]$, we have that $F_i(\vec{y}') \subset F_i(\vec{y})$. To conclude, notice that for every $s_i \in F_i(\vec{y}')$ the utility of the manipulative agent decreases, as all the distances from the facilities have increased after the manipulation, which concludes the proof. □

Unfortunately, not every percentile mechanism is ES: consider the situation represented in Example 1: the position of the facilities are the output of the $\mathcal{PM}_{\vec{v}}$ with $\vec{v} = (0.25, 0.75)$, however, different NE induce different SW values. In what follows, we characterize the set of percentile mechanisms that are ES and study their approximation ratio. Owing to this characterization, we identify the ES percentile mechanism with the lowest approximation ratio. For the sake of simplicity, we start our discussion from the case in which $m = 2$.

## 3.1 MECHANISMS TO PLACE TWO FACILITIES

First, we study the case in which we place two facilities. We denote with $k_1$ and $k_2$ the capacities of the two facilities and assume that $k_1 + k_2 < n$. Without loss of generality, let $k_1 \geq k_2$. First, we show that for every $k_1$ and $k_2$, there exists at least a percentile mechanism that is ES. Moreover, for every $k_1$ and $k_2$, we compute the approximation ratio of every ES percentile mechanism and characterize the mechanism achieving the lowest approximation ratio. In what follows we assume that $v_1 < v_2$, as the case $v_1 = v_2$ is equivalent to the case $m = 1$.

**Theorem 3.** Let $\vec{v} = (v_1, v_2) \in [0, 1]^2$ be a percentile vector and let $\mathcal{PM}_{\vec{v}}$ be its associated percentile mechanism. Let $k_1, k_2 \in \mathbb{N}$ and let $n \in \mathbb{N}$ be such that $k_1 + k_2 < n$ and $\lfloor v_2(n-1) \rfloor - \lfloor v_1(n-1) \rfloor > 1$. Then, we have that $\mathcal{PM}_{\vec{v}}$ is ES if and only if

$$\lfloor v_2(n-1) \rfloor - \lfloor v_1(n-1) \rfloor \geq k_2 + k_1 - 1. \tag{2}$$

*Proof.* First, we show that condition (2) is sufficient to make a percentile mechanism ES. If $\vec{v}$ satisfies (2), there are always $k_1 + k_2$ agents such that $y_1 \leq x_i \leq y_2$, where $y_1$ and $y_2$ are the two facility locations returned by the mechanism. Notice that if $y_1 = y_2$, then the two facilities share the position with $k_1 + k_2$ agents, hence the SW of any NE is equal to $k_1 + k_2$. Assume now that $y_1 < y_2$. Let us then define $r_j$ the minimal values such that $B_{r_j}(y_j) \cap \{x_i\}_{i \in [n]}$ has cardinality larger or equal to $k_j$[1]. Since there are at least $k_1 + k_2$ agents in $[y_1, y_2]$, we have that $r_1 + r_2 \leq |y_1 - y_2|$. According to every NE, agents that do not belong to either $B_{r_1}(y_1)$ or $B_{r_2}(y_2)$ have utility equal to 0. Indeed, if $\vec{s} \in NE(\vec{x}, \mathcal{PM}_{\vec{v}}(\vec{c}))$ is such that $x_i \notin B_{r_1}(y_1) \cap B_{r_2}(y_2)$ gets accommodated by $y_1$, we would have that at least one agent $x_j \in B_{r_1}(y_1)$ has null utility, hence $s_j = 2$. However, if agent $x_j$ can increase its utility by changing its strategy to $s_j = 1$, hence $\vec{s} \notin NE(\vec{x}, \mathcal{PM}_{\vec{v}}(\vec{x}))$, which is a contradiction. By the same argument, we infer that every agent $x_i \in (y_j - r_j, y_j + r_j)$ attains utility $1 - |x_i - y_j|$ according to every NE. Finally, we observe that the set of agents such that $|x_i - y_j| = r_j$ that have non null utility may change according to the specific NE, but the total utility of these

---

[1] Here $B_r(y)$ denotes the ball centered in $y$ of radius $r$.

agents does not change. Thus condition (2) is sufficient to ensure $\mathcal{PM}_{\vec{v}}$ is an ES percentile mechanism.

Lastly, we show that the condition is necessary. Let us assume that $\vec{v}$ does not satisfy the condition (2). For the sake of simplicity, let us denote with $i_1 = \lfloor v_1(n-1) \rfloor + 1$ and $i_2 = \lfloor v_2(n-1) \rfloor + 1$. Let us consider the following instance $x_1 = \cdots = x_{i_1-1} = 0, x_{i_1} = 0.4, x_{i_1+1} = \cdots = x_{i_2-1} = 0.5, x_{i_2} = 0.6$ and $x_j = 0.9$ for all the other indexes $j > i_2$. Notice that, since $\lfloor v_2(n-1) \rfloor - \lfloor v_1(n-1) \rfloor > 1$, there is at least one agent located at $0.5$. Following the same argument used in Example 1, we can show that, depending on the strategy played by the agents at $0.5$, the Social Welfare of the mechanism changes. □

Next, we characterize the approximation ratio of every ES percentile mechanism. Given a percentile vector $\vec{v} \in [0,1]^2$ that satisfies condition (2), we denote with $i_1 = \lfloor v_1(n-1) \rfloor + 1$ and $i_2 = \lfloor v_2(n-1) \rfloor + 1$. Therefore that the mechanism places the facility with capacity $k_1$ at $x_{i_1}$ and the facility with capacity $k_2$ at $x_{i_2}$.

**Theorem 4.** Given $n$, $k_1$, and $k_2$, let $\mathcal{PM}_{\vec{v}}$ an ES percentile mechanism. Then, if $i_1 \geq \lfloor \frac{k_1+1}{2} \rfloor$, we have that

$$ar(\mathcal{PM}_{\vec{v}}) = \frac{k_1 + k_2}{\min\{k_1 + (n - i_2) + 1, \frac{k_1+1}{2} + k_2\}} \quad (3)$$

If $i_1 < \lfloor \frac{k_1+1}{2} \rfloor$ and $i_2 < n - \lfloor \frac{k_2+1}{2} \rfloor$, we have

$$ar(\mathcal{PM}_{\vec{v}}) = \frac{k_1 + k_2}{\min\{k_1 + (n - i_2) + 1, i_1 + k_2\}}.$$

Otherwise, we have

$$ar(\mathcal{PM}_{\vec{v}}) = \frac{k_1 + k_2}{\min\{k_1 + \frac{k_2+1}{2}, k_2 + i_1\}}.$$

*Proof.* We show the result for the case in which $i_1 \geq \lfloor \frac{k_1+1}{2} \rfloor$, the other cases follow by a similar argument and is deferred to the Appendix.

Our argument is as follows: we show that the worst instance for the mechanism is either (i) $x_i = 0$ if $i \in \{1, \ldots, i_1-1\}$, $x_{i_1} = \frac{1}{2}$, and $x_i = 1$ otherwise, or (ii) $x_i = 0$ if $i \in \{1, \ldots, i_2 - 1\}$ and $x_i = 1$ otherwise. Notice that in both cases, the optimal SW is equal to $k_1 + k_2$, which is the maximum SW attainable. Let us show that the worst case instance has the form described in (i) or (ii). Owing to Theorem 3, we have that there are at least $k_1 + k_2$ agents in the interval $[y_1, y_2]$, hence the agents that are accommodated by the facility at $y_j$ are, up to ties, the $k_j$ agents that are closer to $y_j$. Since $i_1 \geq \lfloor \frac{k_1+1}{2} \rfloor$, the total utility of the agent accommodated by the facility at $y_1$ is minimized when all the agents accommodated by $y_1$ are all at the same distance from $y_1$, that is $|x_{i_1-1} - y_1| = |x_{i_1+1} - y_1|$. Given $\lambda \in [0,1]$, let us consider the following instance: $x_1 = \cdots = x_{i_1-1} = 0$, $x_{i_1} = \lambda$, $x_{i_1+1} = \cdots = x_{i_2-1} = 2\lambda$. Let us now consider

the facility located at $y_2$. By the same argument, we have that if $n - i_2$ is larger than $\lfloor \frac{k_2+1}{2} \rfloor$, then, for every $\lambda$, the position $y_2$ that minimizes the utility is $\frac{1}{2} + \lambda$. In this case, the cost of the mechanism is $2 + (1-\lambda)(k_1-1) + (\frac{1}{2}+\lambda)(k_2-1)$. Since $k_1 \geq k_2$, we have that the minimum SW is achieved when $\lambda = \frac{1}{2}$, thus all the agents on the right of $x_{i_1}$ are located at 1, all the agents on the left are located at 0 and $x_{i_1} = \frac{1}{2}$. In this case $SW_{\mathcal{PM}_{\vec{v}}}(\vec{x}) = \frac{k_1+1}{2} + k_2$, while $SW_{opt}(\vec{x}) = k_1 + k_2$, which is the maximum utility achievable and is attained by placing both facilities at 1.

Consider now the case $n - i_2 < \lfloor \frac{k_2+1}{2} \rfloor$. In this case, for every $\lambda$, the worst position for $y_2$ is 1, hence the SW of the mechanism is $i_2 + 1 + (1-\lambda)(k_1-1) + 2\lambda(k_2 - i_2)$, thus, if $2(k_2 - i_2) > (k_1 - 1)$, the SW is minimized when $\lambda = 0$. From a similar computation, we retrieve that $SW_{\mathcal{PM}_{\vec{v}}}(\vec{x}) = k_1 + i_2$, while $SW_{opt}(\vec{x}) = k_1 + k_2$. We conclude the thesis by combining these two cases. □

Consequentially, we characterize the best ES percentile mechanisms given any 2-dimensional vector $\vec{k}$. In particular, we show that the approximation of the best percentile mechanism decreases as $\Delta := n - (k_1 + k_2)$ increases.

**Theorem 5.** Given $n$ and $\vec{k}$, let us define $\Delta = n - (k_1 + k_2)$, then we have that the ES percentile mechanism that achieves the lowest approximation ratio is induced by the percentile vector $\vec{v} = \left( \frac{i_1}{n}, \frac{i_2}{n} \right)$, where $i_1$ and $i_2$ are as follows

(i) $i_1 = \lceil \frac{k_1}{2} \rceil$ and $i_2 = n - \lfloor \frac{k_2}{2} \rfloor$ if $\Delta \geq \lceil \frac{k_1+k_2}{2} \rceil$, in which case $ar(\mathcal{PM}_{\vec{v}}) = \frac{k_1+k_2}{\frac{k_1+1}{2}+k_2}$,

(ii) $i_1 = k_1 - k_2 + \alpha$ and $i_2 = n - \alpha$, where $\alpha = \lceil \frac{\Delta-(k_1-k_2)}{2} \rceil$, if $k_1 - k_2 \leq \Delta \leq \lfloor \frac{k_1+k_2}{2} \rfloor + 1$, in which case $ar(\mathcal{PM}_{\vec{v}}) = \frac{k_1+k_2}{i_1+k_2}$, and

(iii) $i_1 = \Delta + 1$ and $i_2 = n$ otherwise, in which case $ar(\mathcal{PM}_{\vec{v}}) = \frac{k_1+k_2}{\Delta+k_2+1}$.

Notice that the lowest approximation ratio is achieved when $\Delta \geq \lceil \frac{k_1+k_2}{2} \rceil$. Moreover, notice that the smaller the gap between $k_1$ and $k_2$, that is $k_1 - k_2$, the lower the approximation ratio of the best percentile mechanism. In particular, the lowest approximation ratio is attained when $k_1 = k_2$ and $n \geq 3k$, in which case there exists a percentile mechanism whose approximation ratio is $\frac{4}{3+\frac{1}{k}} \sim \frac{4}{3}$.

### 3.2 BEYOND TWO FACILITIES

We now extend our study to the case in which we want to place $m > 2$ facilities. For the sake of simplicity, we consider $m$ facilities that have the same capacity $k$. First, we extend Theorem 3 to this framework.

**Theorem 6.** Let $k$ be the capacity of $m$ facilities. Moreover, let $\vec{v}$ be a percentile vector such that $v_1 < v_2 < \cdots < v_m$ so that $\vec{v}$ does not possess two equal entries and let $\mathcal{PM}_{\vec{v}}$ be

its associated percentile mechanism. Moreover, assume that $\lfloor v_{j+1}(n-1) \rfloor - \lfloor v_j(n-1) \rfloor > 1$ for every $j \in [m-1]$. Then $\mathcal{PM}_{\vec{v}}$ is ES if and only if the following system of inequalities is satisfied

$$\begin{cases} \lfloor v_2(n-1) \rfloor - \lfloor v_1(n-1) \rfloor \geq 2k-1 \\ \dots \\ \lfloor v_m(n-1) \rfloor - \lfloor v_{m-1}(n-1) \rfloor \geq 2k-1 \end{cases} \quad . \quad (4)$$

The proof of Theorem 6 follows an argument similar to the one used to prove Theorem 3, so we defer it to the Appendix. The set of inequalities (4) allows us to characterize the vectors $\vec{v}$ that induce an ES percentile mechanism $\mathcal{PM}_{\vec{v}}$ depending on the capacity $k$. Notice that system (4) is not feasible if $k > \frac{n+m}{2m}$ or, equivalently, $n < (2k-1)m$. Indeed, by summing all the inequalities in (4), we have that

$$\lfloor v_m(n-1) \rfloor - \lfloor v_1(n-1) \rfloor \geq (2k-1)m.$$

Since $n \geq \lfloor v_m(n-1) \rfloor - \lfloor v_1(n-1) \rfloor$, we must indeed have that $n \geq (2k-1)m$. Although when $n < (2k-1)m$ it is impossible to define an ES percentile mechanism that places $m$ facilities at $m$ different locations, it is possible to define an ES percentile mechanism that places all the facilities at one or two different locations. To keep the discussion on track, we first study the case in which system (4) admits a solution and defer the pathologic case to a dedicated section.

### 3.2.1 Case $n \geq (2k-1)m$.

In this case, it is possible to select an ES and absolutely truthful percentile mechanism that places the $m$ facilities at $m$ different positions among the agents' reports.

**Theorem 7.** If $k < \frac{n+m}{2m}$, then given a ES $\mathcal{PM}_{\vec{v}}$, we have

$$ar(\mathcal{PM}_{\vec{v}}) = \begin{cases} \frac{mk}{(m-\frac{1}{2})k+\frac{1}{2}} & \text{if } i_1, n-i_m \geq \lfloor \frac{k+1}{2} \rfloor \\ \frac{mk}{(m-1)k+\min\{i_1,n-i_m\}} & \text{otherwise} \end{cases}$$

where $i_1 = \lfloor v_1(n-1) \rfloor + 1$ and $i_m = \lfloor v_m(n-1) \rfloor + 1$.

*Proof.* The case in which $i_1, n-i_m \geq \lfloor \frac{k+1}{2} \rfloor$ follows by the same argument adopted in the proof of Theorem 4. Indeed, by definition of the mechanism, the SW of the mechanism is minimized when each facility $y_j = x_{\lfloor v_j(n-1) \rfloor + 1}$ is such that $|y_j - x_{\lfloor v_j(n-1) \rfloor}| = |y_j - x_{\lfloor v_j(n-1) \rfloor + 2}|$. Hence the mechanism achieves the minimal SW when $x_{\lfloor v_j(n-1) \rfloor + 1} = \frac{2j-1}{2m}$ for every $j \in [m]$ and $x_i = \frac{l}{m}$ if $\lfloor v_l(n-1) \rfloor + 1 < i < \lfloor v_{l+1}(n-1) \rfloor + 1$ where $l = 0, 1, \dots, m$, $v_0 = 0$ and $v_{m+1} = 1$. On such instance the SW of the mechanism is $(m - \frac{1}{2})k + \frac{1}{2}$. Notice the mechanism achieves the same SW on the instance $\vec{x}_O$ defined as $(x_O)_i = 0$ for every $i \leq \lfloor v_1(n-1) \rfloor + 1$, and $(x_O)_i = 1$ otherwise. To conclude, we observe that the optimal SW on instance $\vec{x}_O$ is $mk$.

The case in which $\min\{i_1, n-i_m\} \leq \lfloor \frac{k+1}{2} \rfloor$ follows a similar argument and it is deferred to the Appendix. $\square$

In particular, for every given the capacity $k$ and number of facility $m$, it is possible to detect the best possible ES and absolutely truthful percentile mechanism.

**Theorem 8.** Given $k$, $m$, and $n$, let us define $\alpha = \lfloor \frac{(n-2k(m-1)+1)}{2} \rfloor$. The vector $\vec{v}$ where $v_j = \frac{\alpha+(2k-1)(j-1)}{n}$ for $j \in [m]$ induces the ES percentile mechanism with the lowest approximation ratio. In particular, if $n \geq 2km$, the approximation ratio of $\mathcal{PM}_{\vec{v}}$ is less than $1 + \frac{1}{2m-1}$.

Notice that, if $n \geq 2km$, the approximation ratio of the best percentile mechanism decreases as the number of facilities increases. Noticeably, when $m$ goes to infinity, the approximation ratio goes to 1.

### 3.2.2 Case $n < (2k-1)m$.

We now consider the case in which the number of agent is too small and thus Theorem 6 does not hold. In this case, it is possible to circumvent Theorem 6, by considering an percentile mechanism that places all the facilities at either one or two locations, that is the percentile mechanisms whose associated vector $\vec{v}$ has at most two different entries. When more than one facility is placed at the same location, we considered them as a unique facility whose capacity is the sum of all the facilities placed at the common location.

**Two different entries.** When the mechanism places the facilities at two different locations, we can use the results proposed in Section 3.1. Indeed, owing to Theorem 5, we know that the approximation ratio becomes lower as the difference in capacity between facilities is smaller. For this reason, we consider a mechanism that splits the facilities as evenly as possible.

**Mechanism 1** (All-aside mechanism). Let $k$ be the capacity of $m$ facilities and let $a, b \in \mathbb{N}$ be such that $a + 2mk \leq b \leq n$. Given in input a vector $\vec{x} \in [0,1]^n$, the All-aside mechanism associated with $a$ and $b$, namely $AS_{a,b}$, places $\lceil \frac{m}{2} \rceil$ facilities at $x_a$ and $\lfloor \frac{m}{2} \rfloor$ facilities at $x_b$.

Owing to Theorem 3, the All-aside is absolutely truthful and ES. Moreover, we can extend Theorem 4 to this case.

**Theorem 9.** The approximation ratio of every $AS_{a,b}$ is determined by Theorem 4 by setting $k_1 = \lceil \frac{m}{2} \rceil k$, $k_2 = \lfloor \frac{m}{2} \rfloor k$, $i_1 = a$, and $i_2 = b$.

**One different entry.** Lastly, we consider the case in which the mechanism places all the facilities at one place, hence the percentile vector $\vec{v} = (v, v, \dots, v)$ for a $v \in [0,1]$. In this case, we have that most of the results presented in (Aziz et al. [2020b]) extend trivially, hence every $\vec{v} = (v, v, \dots, v)$ induces an absolutely truthful and ES mechanism. Moreover, the best percentile vector is $\vec{m} = (0.5, 0.5, \dots, 0.5)$. In our case, however, the approximation ratio guarantees are worse

than the one presented in (Aziz et al. [2020b]). Indeed, since in our case the capacity can be split at $m$ different locations, the optimal solution has a further degree of freedom that heightens the approximation ratio of the mechanism.

**Theorem 10.** Let $k > 1$ be the capacity of the facilities and fix $\vec{v} = (0.5, 0.5, \ldots, 0.5)$. If $n \le (m+1)k$, we have that

$$ar(\mathcal{PM}_{\vec{v}}) = \frac{2(m-1)k + (n - (m-1)k) + 1}{mk + 1}.$$

Otherwise, $ar(\mathcal{PM}_{\vec{v}}) = \frac{2mk}{mk+1} = 2 - \frac{2}{mk+1}$.

Notice that the lowest approximation ratio occurs when $n = km + 1$, in which case $ar(\mathcal{PM}_{\vec{v}}) = 2 - \frac{k}{mk+1}$. Thus, as the number of facilities increases, we attain an approximation ratio that converges to 2.

## 4 EXPERIMENTAL RESULTS

In this section, we complement our theoretical study of the $m$-CFLP with scarce resources by running several numerical experiments. In particular, we evaluate the Bayesian approximation ratio of the percentile mechanisms identified by Theorem 5. The Bayesian approximation ratio measures how close the expected SW induced by the mechanism and the expected optimal SW are when the agents' positions are samples drawn from a probability distribution, (Hartline and Lucier [2010]). Therefore, the Bayesian approximation of $\mathcal{PM}_{\vec{v}}$ is

$$B_{ar}(\mathcal{PM}_{\vec{v}}) := \frac{\mathbb{E}_{\vec{X} \sim \mu}[SW_{\mathcal{PM}_{\vec{v}}}(\vec{X})]}{\mathbb{E}_{\vec{X} \sim \mu}[SW_{opt}(\vec{X})]}, \tag{5}$$

where $\vec{X}$ is a $n$ dimensional random vector distributed according to $\mu$. Our aim is to show that percentile mechanisms that are optimal according to the worst-case analysis, namely $\mathcal{PM}_{best}$ (see Theorem 5), are optimal or quasi-optimal from a Bayesian perspective as well. For this reason, we run two tests: (i) first, we assess to what extent the Bayesian approximation ratio depends on the percentile vector inducing the percentile mechanism when all the agents are independent and identically distributed (i.i.d.). In particular, we compare $\mathcal{PM}_{best}$ with $\mathcal{PM}_{(0,1)}$, that is the percentile mechanism induced by $\vec{v} = (0, 1)$. (ii) Secondly, we assess the Bayesian approximation ratio of $\mathcal{PM}_{best}$ when diverse agents within the populations follow distinct distributions. This examination helps determine the suitability of $\mathcal{PM}_{best}$ for addressing problems involving non-identically distributed agents.

We run our experiments for different distributions $\mu$ and different capacity vectors $\vec{k}$ in order to provide a comprehensive view. Moreover, since the highest approximation ratio is attained when $m = 2$, we only consider cases in which we need to place two facilities. Due to space limits, part of the results are deferred to the Appendix.

Throughout our experiments, we sample the agents' positions from three different probability distributions supported over $[0, 1]$: (i) the uniform distribution, namely $\mathcal{U}$ whose density is equal to 1 over a $[0, 1]$, (ii) the triangular distributions of parameter $c \in (0, 1)$, namely $\mathcal{T}$, whose density is equal to $2(1 - x)$ over a $[0, 1]$, and (iii) the Beta distributions of parameters $\alpha, \beta > 0$, namely $\mathcal{B}(\alpha, \beta)$ whose density is equal to $Cx^{\alpha-1}(1 - x)^{\beta-1}$ over a $[0, 1]$, where $C$ is a normalizing constant. We consider different capacity vectors $\vec{k}$. Specifically, we consider balanced capacities $\vec{k} = (k, k)$ and unbalanced capacities $\vec{k} = (k_1, k_2), k_1 > k_2$. For the case of balanced capacities, we consider $k = \alpha n$, where $\alpha = 0.1, 0.2, 0.3$, and $0.4$. For the case of unbalanced capacities, we consider the slightly unbalanced capacities i.e. $\vec{k} = (0.4n, 0.3n)$, and highly unbalanced capacities i.e. $\vec{k} = (0.6n, 0.2n), (0.7n, 0.1n)$. Lastly, for every instance $\vec{x}$, we do not compute the optimal SW, but rather an upper bound to that quantity. Indeed, the optimal position of the facilities can be any couple of points in $[0, 1]$. Furthermore, to select the optimal facility location we must compute all the NE of every possible facility location and select the one achieving the highest SW. For these reasons, we consider an easier to compute upper bound that is obtained by considering the maximum SW achievable when the mechanism forces agents to use a specific facility, that is

$$SW_{UB}(\vec{x}) := \sup_{y_1, y_2 \in [0,1]} \sup_{\pi \in \Pi} \sum_{i \in [n_u]} \sum_{j \in [2]} (1 - |x_i - y_j|)\pi_{i,j}$$

where $\Pi$ is the set containing all the matching $\pi \subset [n] \times [2]$ such that (i) every $i \in [n]$ has degree that is equal or lower than 1 and (ii) every $j \in [2]$ has degree equal to $k_j$. This quantity is easy to compute, as the set of optimal positions for the facilities coincides with the positions of the agents.

**Experiment results – Comparing different percentile mechanisms.** In this experiment we want to assess to what degree the percentile vector affects the performances of the percentile mechanisms it induces. For this reason, we compare the empiric Bayesian approximation ratio of the best percentile mechanism $\mathcal{PM}_{best}$ (identified via Theorem 5), with the performances of a mechanism that places the facilities at the extreme agents' positions. That is, the percentile mechanism induced by the vector $\vec{w} = (0, 1)$, namely $\mathcal{PM}_{(0,1)}$. We first consider the case of balanced capacities $\vec{k} = (k, k)$. Figure 1 shows the average and the 95% confidence interval (CI) of Bayesian approximation ratio for $n = 10, 20, 30, 40, 50$ when the agents are distributed according to $\mathcal{U}$ and $\mathcal{T}$. Each average is computed over 500 instances. We observe that the percentile mechanism identified in Theorem 5 achieves the a Bayesian approximation ratio that is lower than the one obtained by $\mathcal{PM}_{(0,1)}$ for every value of $n$. Moreover, the Bayesian approximation ratio of $\mathcal{PM}_{(0,1)}$ consistently increases as the number of agents increases, while the Bayesian approximation ratio of $\mathcal{PM}_{best}$ remains constant regardless of $n$. Figure 2 shows the average

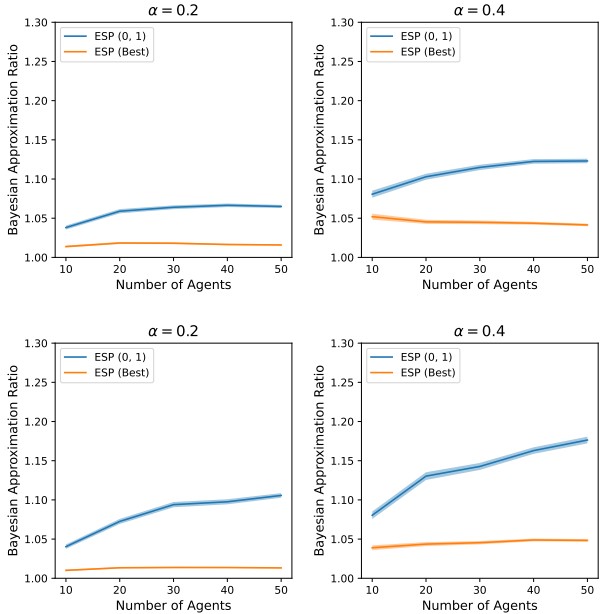

Figure 1: The Bayesian approximation ratio of $\mathcal{PM}_{best}$ and $\mathcal{PM}_{\vec{w}}$ in the balanced case, i.e. $k_1 = k_2 = \alpha n$ for $n = 10, 20, \ldots, 50$. Every column contains the results for different vector $\vec{k}$. The first row contains the results for the Uniform distribution $\mathcal{U}$, while the second row the results for the triangular distribution $\mathcal{T}$.

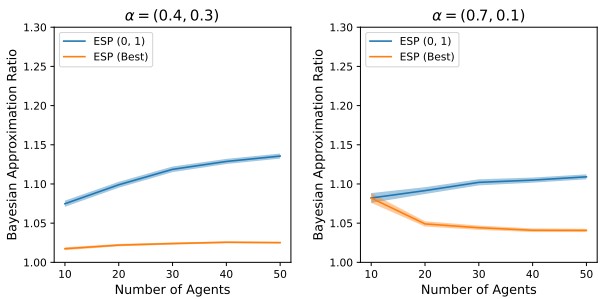

Figure 2: The Bayesian approximation ratio of $\mathcal{PM}_{best}$ and $\mathcal{PM}_{\vec{w}}$ when the agents are distributed according to $\mathcal{T}$ and the facilities are unbalanced, i.e. $k_1 = \alpha_1 n \neq k_2 = \alpha_2 n$ for $n = 10, 20, \ldots, 50$. Every column contains the results for different vector $\vec{k}$.

and the $95\%$ confidence interval (CI) of Bayesian approximation ratio for $n = 10, 20, 30, 40, 50$ when the capacities of the two facilities are not balanced, and the agents are distributed according to $\mathcal{T}$. More specifically, we consider $\vec{k} = (0.4n, 0.3n)$ and $(0.7n, 0.1n)$. Again, we observe that the percentile mechanism identified by Theorem 5 has a lower and more stable Bayesian approximation ratio, highlighting how choosing a percentile vector affects the average performances of the mechanism.

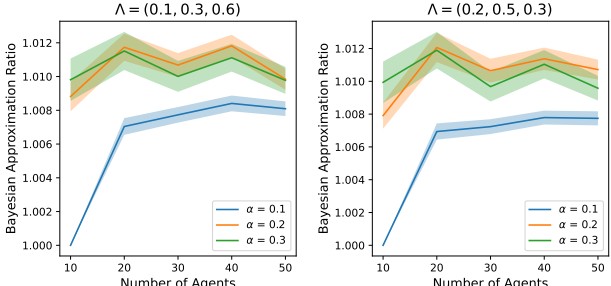

Figure 3: The Bayesian approximation ratio of $\mathcal{PM}_{best}$ for a non i.d. population in the balanced, i.e. $k_1 = k_2 = \alpha n$ with $\alpha = 0.1, 0.2, 0.3$, and for $n = 10, 20, \ldots, 50$. Every column contains the results for different $\Lambda$.

**Experiment results – Bayesian approximation ratio for non i.d. populations.** In this test, we empirically evaluate the Bayesian approximation ratio of the percentile mechanisms identified by Theorem 5 when agents are not identically distributed. In particular, we consider the case in which each agent of the population is distributed according to $\mathcal{U}$, $\mathcal{T}$, and $\mathcal{B}(5,5)$. Every instance is identified by the percentages of agents following each distributions, hence we set $\Lambda = (\lambda_U, \lambda_B, \lambda_T)$, where $\lambda_U = \frac{n_U}{n}$, $\lambda_B = \frac{n_B}{n}$, and $\lambda_T = \frac{n_T}{n}$, $n$ is the total number of agents, and $n_U$, $n_B$, and $n_T$ are the number of agents following the uniform, Beta, and Triangular distribution, respectively.

In Figure 3, we report two instances with two different vectors $\Lambda$. We consider the case in which the capacities of the facilities are balanced, that $k_1 = k_2 = \lfloor \alpha n \rfloor$, where $\alpha = 0.1$, $0.2$, and $0.3$. From our experiments we observe that the percentile mechanism achieves an almost optimal Bayesian approximation ratio (peaking at $1.01$), that it is constant regardless of $n$, and that the CI is small (around $0.003$). Our experiments confirm that the best percentile mechanism according to the worst-case analysis behave almost optimally in a Bayesian framework.

## 5 CONCLUSION AND FUTURE WORKS

In this paper, we studied the mechanism design aspects of the $m$-CFLP under the assumption that the total capacity of the facility is smaller than the number of agents to accommodate. We assume that, after the position of the facility is fixed, the agents compete in a First-Come-First-Served (FCFS) game to gain access to the facilities. Our main contribution consist in studying the case in which $m \geq 2$, which was left as an open questions in the paper introducing the problem (Aziz et al. [2020b]). Our approach emphasizes the significance of absolutely truthful mechanisms, which prevent agents from benefiting regardless of their strategy in the FCFS game, and ES mechanisms, whose SW remains independent of the FCFS game equilibrium. We show that

the percentile mechanisms (Sui et al. [2013]) are absolutely truthful and characterize under which conditions they are ES. We show that ES percentile mechanisms achieve bounded approximation ratio for every $m > 1$ and characterize the best percentile vector as a function of $n$, $k_1$, and $k_2$. Interestingly, if $n > (2k-1)m$, the approximation ratio of the best percentile mechanism $1 + \frac{1}{2m-1}$, i.e. is asymptotically optimal with respect to the number of facilities. Lastly, we run extensive numerical results to study the performances of the percentile mechanism from a Bayesian perspective.

In our future works, we aim at extending this problem to the case in which the agents are distributed to higher dimensional spaces or graphs. Another interesting research avenue is to study how changing the preferences of the agents affects the performances of the mechanisms. Finally, it would be interesting to study the asymptotic Bayesian approximation ratio as done in (Auricchio et al. [2024b]) and beyond worst-case analysis proposed in (Deng et al. [2022]) to complement and strengthen our experimental results.

## Acknowledgements

Jie Zhang was partially supported by a Leverhulme Trust Research Project Grant (2021 – 2024) and the EPSRC grant (EP/W014912/1).

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

## APPENDIX

In this appendix, we report the proofs and the experimental results missing from the main body of the paper.

## A  MISSING PROOFS

*Proof of Theorem 1.* Let $\vec{x}$ be the vector containing the position of the agents and let $\vec{y}$ be the position of the facilities. We denote with $k_j$ the capacity of the facility located at $y_j$ for every $j \in [m]$. In what follows, we assume that the set of agents has an inner ordering that decides how to break ties.

Let us define $\mathcal{D}$ the set containing all the distances agents to facility, that is $\mathcal{D} = \{|x_i - y_j|\}_{i\in[n],j\in[m]}$.

Let $\vec{c} \in \mathbb{R}^m$ be the null vector, that is $c_j = 0$ for every $j \in [m]$. We now construct a Nash Equilibrium through the following iterative routine.

1.  Let $d$ be the minimum of the elements in $\mathcal{D}$. Up to a tie, there exist a couple $(i_1, j_1) \in [n] \times [m]$ such that $d = |x_{i_1} - y_{j_1}|$. We set $c_{j_1} = c_{j_1} + 1$, $s_{i_1} = j_1$, and remove all the elements of the form $|x_{i_1} - y_j|$ from $\mathcal{D}$. Then, if $c_{j_1} = k_{j_1}$, we remove from $\mathcal{D}$ all the elements of the form $|x_i - y_{j_1}|$.
2.  We repeat the routine of point $(1)$ until $\mathcal{D}$ becomes empty.
3.  If $s_i = 0$ for some $i \in [n]$, we set them to be equal to $1$.

Since $\mathcal{D}$ is discrete, the routine terminates in finite number of iterations and the output is a vector containing a set of agents' pure strategies.

We now show that the output of the routine $\vec{s}$ is a Nash Equilibrium by proving that no agent $i$ can increase its payoff by deviating from playing $s_i$. Toward a contradiction, assume that an agent $i$ can increase its payoff by playing $s_i'$ rather than $s_i$. By definition of $s_i$, we have that if $|x_i - y_{s_i'}| < |x_i - y_{s_i}|$, then there are at least $k_{s_i'}$ agents that are closer to $y_{s_i'}$ or that have a higher priority order than agent $i$ and play strategy $s_i'$. Thus the agent cannot gain a benefit from deviating from $s_i$, which proves that $\vec{s}$ is a pure Nash Equilibrium. $\square$

*Proof of Theorem 4.* To complete the proof, we need to consider the case in which $i_1 < \lfloor \frac{k_1+1}{2} \rfloor$. First, we consider the case in which $i_2 < n - \lfloor \frac{k_2+1}{2} \rfloor$. By the same argument used to prove the case in which $i_1 \geq \lfloor \frac{k_1+1}{2} \rfloor$, we have that the worst case instance in this case is

$$x_i = \begin{cases} x_i = 0 & \text{if} \quad i = 1, \ldots, i_1, \\ x_i = \lambda & \text{if} \quad i = i_1 + 1, \ldots, i_2 - 1, \\ x_i = 1 & \text{otherwise.} \end{cases}$$

for some $\lambda \in [0, 1]$, since the SW of the mechanism is minimized when the $i_1$-th and $i_2$-th agents are at the extremes of the interval. For any value of $\lambda$, the SW of the mechanism is then

$$SW(\vec{x}) = i_1 + (n - i_2) + (1 - \lambda)(k_1 - i_1) + \lambda(k_2 - (n - i_2)).$$

Since $SW(\vec{x})$ is linear in $\lambda$, we have that the minimum is achieved at either $\lambda = 0$ or $\lambda = 1$. Thus the minimal SW achievable is

$$\min\{k_1 + (n - i_2), k_2 + i_1\}.$$

Since in both cases we have that the optimal SW is $k_1 + k_2$, we conclude the thesis for this specific case.

Lastly, we consider the case in which $n - i_2 \geq \lfloor \frac{k_2+1}{2} \rfloor$. In this case, the worst case instance places the first $i_1$ agents on the extreme left side, while places $y_2$ in between two clusters of agents. Therefore we consider the following instance

$$x_i = \begin{cases} x_i = 0 & \text{if} \quad i = 1, \ldots, i_1, \\ x_i = \lambda & \text{if} \quad i = i_1 + 1, \ldots, i_2 - 1, \\ x_{i_2} = \frac{\lambda+1}{2} \\ x_i = 1 & \text{otherwise.} \end{cases}$$

The SW induced by the mechanism is then

$$SW(\vec{x}) = i_1 + 1 + (1 - \lambda)(k_1 - i_1) + \frac{1 + \lambda}{2}(k_2 - 1).$$

Again, since the SW is linear in $\lambda$, we have that the minimium is attained at either $\lambda = 0$ or $\lambda = 1$. Then the minimum SW achievable by the mechanism is

$$\min\left\{k_1 + \frac{(k_2+1)}{2}, k_2 + i_1\right\}.$$

To conclude notice that in both cases, the SW attained by the optimal solution is $k_1 + k_2$. $\qquad\square$

*Proof of Theorem 5.* When $\Delta \geq \lceil \frac{k_1+k_2}{2} \rceil$, the indexes $i_1 = \lceil \frac{k_1}{2} \rceil$ and $i_2 = n - \lfloor \frac{k_2}{2} \rfloor$ are well defined. Owing to Theorem 3 and by definition of $\Delta$, we have that $\mathcal{PM}_{\vec{v}}$ is ES. Finally, from Theorem 4, we infer that $ar(\mathcal{PM}_{\vec{v}}) = \frac{k_1+k_2}{\frac{k_1+1}{2}+k_2}$, which is the smallest approximation ratio achievable by an ES percentile mechanism.

To conclude the proof, we need to show that the points (ii) and (iii) hold. We do that by carefully tuning $i_1$ and $i_2$. For the sake of simplicity, we consider $i_1$ and $i_2$ to be rationals, to retrieve the real integer indexes, it suffices to take the floor or the ceil functions of the quantities we retrieve.

Let us consider the case (ii), that is $k_1 - k_2 \leq \Delta \leq \lfloor \frac{k_1+k_2}{2} \rfloor + 1$. Owing to Theorem 4, we retrieve the best values $i_1$ and $i_2$ by maximizing the quantity

$$\min\{k_1 + (n - i_2), i_1 + k_2\}.$$

Thus, we look for $i_1$ and $i_2$ such that

$$k_1 + (n - i_2) = i_1 + k_2,$$

subject to the constraint

$$n - i_2 + i_1 = \Delta,$$

since, owing to Theorem 2, $k_1 + k_2$ agents must lay between $x_{i_1}$ and $x_{i_2}$. By a simple computation, we have that

$$n - i_2 = \frac{k_2 - k_1 + \Delta}{2},$$

thus $i_1 = \frac{\Delta - (k_2 - k_1)}{2} = k_1 - k_2 + \frac{\Delta - (k_2 - k_1)}{2}$ and $i_2 = n - \frac{k_2 - k_1 + \Delta}{2}$, which concludes the proof of case (ii).

Lastly, we consider case (iii). In this case, we have that $\Delta < k_1 - k_2$, thus we have

$$k_2 + i_1 - k_1 - (n - i_2) = i_2 - n + i_1 + k_2 - k_1 \leq \Delta + k_2 - k_1 < 0,$$

since $i_2 - n + i_1 < n - i_2 + i_1 \leq \Delta$. Thus the minimum SW attainable by the mechanism is $i_1 + k_2$, therefore, to maximize the minimum achievable SW, we need to set $i_1 = \Delta$ and $i_2 = n$, which concludes the proof. $\qquad\square$

*Proof of Theorem 6.* The proof follows by the same argument used to prove Theorem 3. Indeed, by condition (4) for every $j \in [m]$ we have that at least $k_j + k_{j+1}$ agents are located between $y_j$ and $y_{j+1}$, thus the Social Welfare generated by the facilities at $y_j$ and $y_{j+1}$ does not depend on the specific Nash equilibrium. To conclude the proof, it suffices to apply this argument to each couple of facilities $(y_j, y_{j+1})$.

$\qquad\square$

*Proof of Theorem 7.* To conclude the proof, we need to consider the case in which either $i_1$ or $n - i_m$ are lower than $\lfloor \frac{k+1}{2} \rfloor$.

Since the other case is symmetric, we restrict our analysis to the case in which $i_1 \leq n - i_2$. Again, since $i_1, n - i_m \leq \lfloor \frac{k+1}{2} \rfloor$, we have that the worst case instance places the first $i_1$ agents at 0 and the last $n - i_m + 1$ at 1. Since every facility has the same capacity, we have that the worst case instance has the following form

$$x_i = \begin{cases} 0 & \text{if } i = 1, \ldots, i_1, \\ \delta_1 & \text{if } i = i_1 + 1, \ldots, i_2 - 1, \\ \delta_1 + \frac{1 - \delta_1 - \delta_2}{2(m-2)} & \text{if } i = i_2, \\ \delta_1 + 2\frac{1 - \delta_1 - \delta_2}{2(m-2)} & \text{if } i = i_2 + 1, \ldots, i_3 - 1, \\ \delta_1 + 3\frac{1 - \delta_1 - \delta_2}{2(m-2)} & \text{if } i = i_3, \\ \delta_1 + 4\frac{1 - \delta_1 - \delta_2}{2(m-2)} & \text{if } i = i_3 + 1, \ldots, i_4 - 1, \\ \ldots \\ 1 - \delta_2 & \text{if } i = i_{m-1} + 1, \ldots, i_m - 1, \\ 1 & \text{otherwise} \end{cases}$$

where $\delta_1, \delta_2 \geq 0$ and such that $\delta_1 + \delta_2 \leq 1$. The SW of the mechanism on this instance is

$$SW(\vec{x}) = i_1 + (n - i_2) + m - 2 + (k - i_1)(1 - \delta_1) + \sum_{i=2}^{m-2}\left( (k-1)\left(\frac{m - 3 + \delta_1 + \delta_2}{m - 2}\right)\right) + (k - (n - i_m))(1 - \delta_2).$$

Again, this quantity is linear in $\delta_1$ and $\delta_2$, thus it is minimized when $\delta_1, \delta_2 \in \{0, 1\}$ By plugging in the possible combinations, we infer that the minimum is achieved when $\delta_1 = 1$ and $\delta_2 = 0$ since $i_1 \leq n - i_m$. $\qquad\square$

*Proof of Theorem 8.* Owing to Theorem 7, the approximation ratio is lower when $\min\{i_1, n - i_m\}$ is maximized, thus when $i_1 = n - i_2$. Thus the best mechanism places the first and last facility at $x_\ell$ and $x_{n-\ell}$, where $\ell$ is a suitable integer. Since $i_1 + n - i_2 = n - 2k(m - 1) + 1$, we complete the first half of the proof.

Notice that, if $i_1$ or $i_2$ is less than $\lfloor\frac{k+1}{2}\rfloor$, then we have that

$$\min\{i_1, i_2\} \leq \lfloor\frac{k+1}{2}\rfloor.$$

Therefore,

$$\left(m - \frac{1}{2}\right)k + \frac{1}{2} - (m - 1)k - \min\{i_1, i_2\} \geq \frac{k}{2} + \frac{1}{2} - \lfloor\frac{k+1}{2}\rfloor \geq 0,$$

thus the approximation ratio of the mechanism is smaller when $i_1, i_2 \geq \lfloor\frac{k+1}{2}\rfloor$. Moreover, in this case, the approximation ratio does not depend on the specific $\vec{v}$, thus any ES percentile mechanism whose $\vec{v}$ is such that $i_1, i_2 \geq \lfloor\frac{k+1}{2}\rfloor$ achieves the minimum approximation ratio. Notice that, by definition, the vector $\vec{v}$ where $v_j = \frac{\alpha + (2k-1)(j-1)}{n}$ for $j \in [m]$ where $\alpha = \lfloor\frac{(n - 2k(m-1)+1)}{2}\rfloor$ is such that $i_1, i_2 \geq \lfloor\frac{k+1}{2}\rfloor$. Moreover, owing to Theorem 2, it is also ES, hence it achieves the minimal approximation ratio.

Lastly, notice that

$$\frac{mk}{(m - \frac{1}{2})k + \frac{1}{2}} \leq \frac{mk}{(m - \frac{1}{2})k} = \frac{(m - \frac{1}{2})k + \frac{k}{2}}{(m - \frac{1}{2})k} = 1 + \frac{1}{2m - 1},$$

which concludes the proof. $\qquad\square$

*Proof of Theorem 9.* It follows directly from Theorem 4. Indeed, it suffices to prove that even if we have $m$ facilities to locate, the optimal SW we can obtain by locating $m$ facilities with capacity $l$ is the same as locating two facilities with capacity $\lceil\frac{m}{2}\rceil k$ and $\lfloor\frac{m}{2}\rfloor k$. Since the worst case instance of any $\mathcal{PM}_{\vec{v}}$ with $\vec{v} \in [0,1]^2$ places $i_1$ agents 0 and the others at 1, the optimal SW remains $mk$ even though we locate $m$ facilities separately. $\qquad\square$

*Proof of Theorem 10.* By definition of $\vec{v} = (0.5, 0.5, \ldots, 0.5)$ and $\mathcal{PM}_{\vec{v}}$, we have that for every input $\vec{x} \in [0,1]^n$ the facility is placed at $\lfloor\frac{n+1}{2}\rfloor$. The number of agents on the left of $y_1$ and the number of agents on the right of $y_1$ is the same, hence the SW of the mechanism is minimizes when $x_i = 0$ when $i < \lfloor\frac{n+1}{2}\rfloor$, $x_{\lfloor\frac{n+1}{2}\rfloor} = \frac{1}{2}$, and $x_i = 1$ otherwise. The SW of the mechanism is $\frac{mk+1}{2}$.

If $n \leq (m+1)k$, the optimal SW on the instance is $(m - 1)k + \frac{n - (m-1)k}{2} + \frac{1}{2}$. Indeed, we can locate $m - 1$ facilities at either 0 or 1 that only accommodate the agents at 0 and 1. The total combined utility of the agents accommodated by these $m - 1$ facilities is $(m - 1)k$. Since the agents are divided evenly among 0 and 1, the maximum utility attainable by the last facility is at most $\frac{n - (m-1)k}{2} + \frac{1}{2}$. Therefore the total utility of the optimal SW is $(m - 1)k + \frac{n - (m-1)k}{2} + \frac{1}{2}$.

If $n > (m+1)k$, the optimal SW on this instance is $mk$, and it is attained when $\lfloor\frac{m}{2}\rfloor$ facilities are placed at 0 and the others at 1. To conclude the thesis it suffices to take the ratio of the optimal SW and the SW of the mechanism. $\qquad\square$

# B  ADDITIONAL EXPERIMENTAL RESULTS

In this section, we report the experimental results missing from the main body of the paper.

In Table 4, we report all our results for the case in which the facilities have balanced capacity, that is $k_1 = k_2$.

In Table 4, we report all our results for the case in which the facilities have unbalanced capacity, that is $k_1 > k_2$.

In Table 6, we report all our experiments non identical for different values of $\Lambda$.

We observe no major changes across all the different cases we considered.

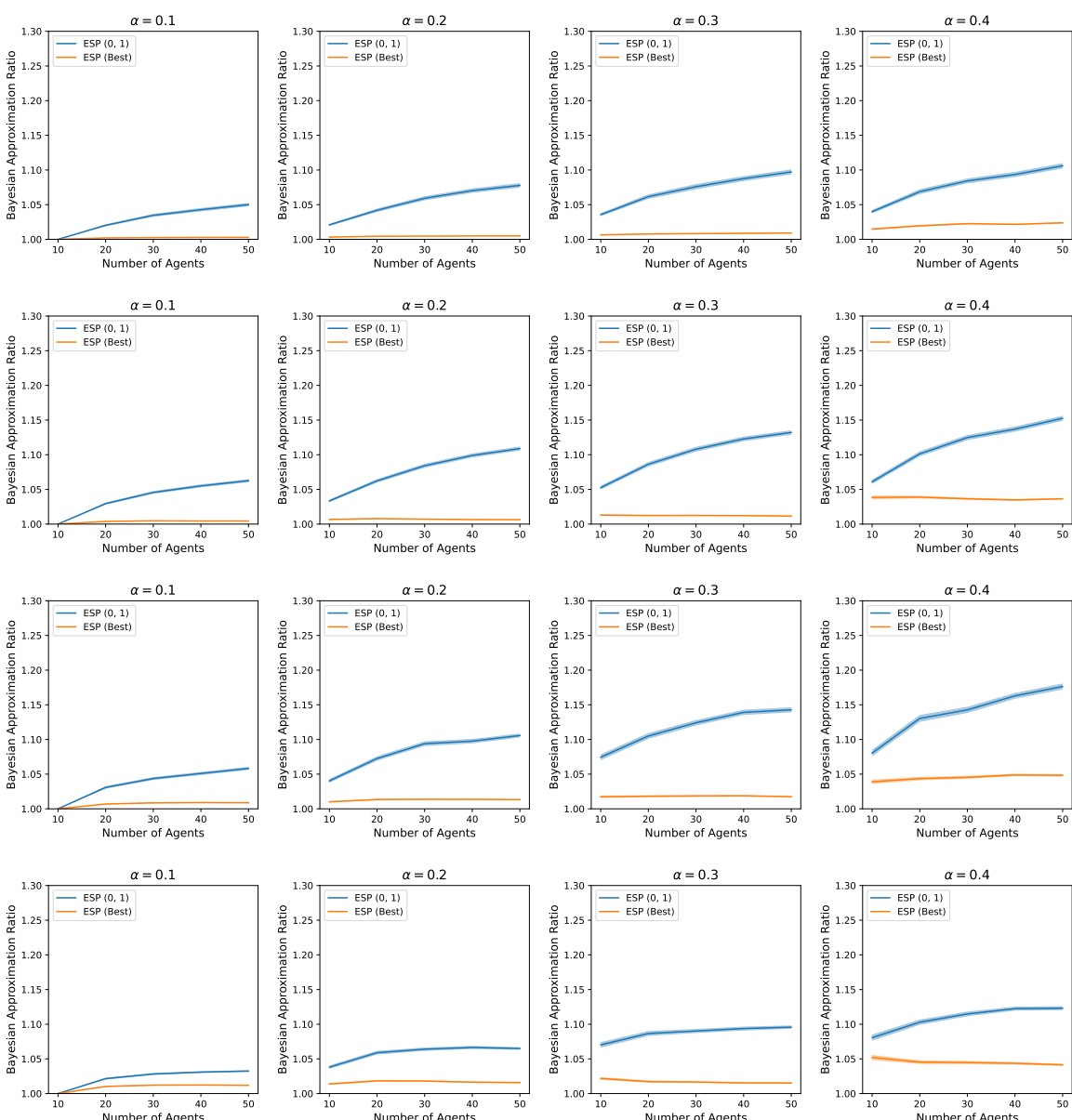

Figure 4: The Bayesian approximation ratio of $\mathcal{PM}_{best}$ and $\mathcal{PM}_{\vec{w}}$ in the balanced case, i.e. $k_1 = k_2$ for $n = 10, 20, \ldots, 50$. Every column contains the results for different vector $\vec{k}$. The first and second row contains the results for the Beta distribution. In the first row, we consider an asymmetric Beta distribution, that is $\mathcal{B}(1, 9)$; in the second row a symmetric Beta, that is $\mathcal{B}(5, 5)$. The third row contains the results for the triangular distribution $\mathcal{T}$. The last row contains the results for the Uniform distribution $\mathcal{U}$.

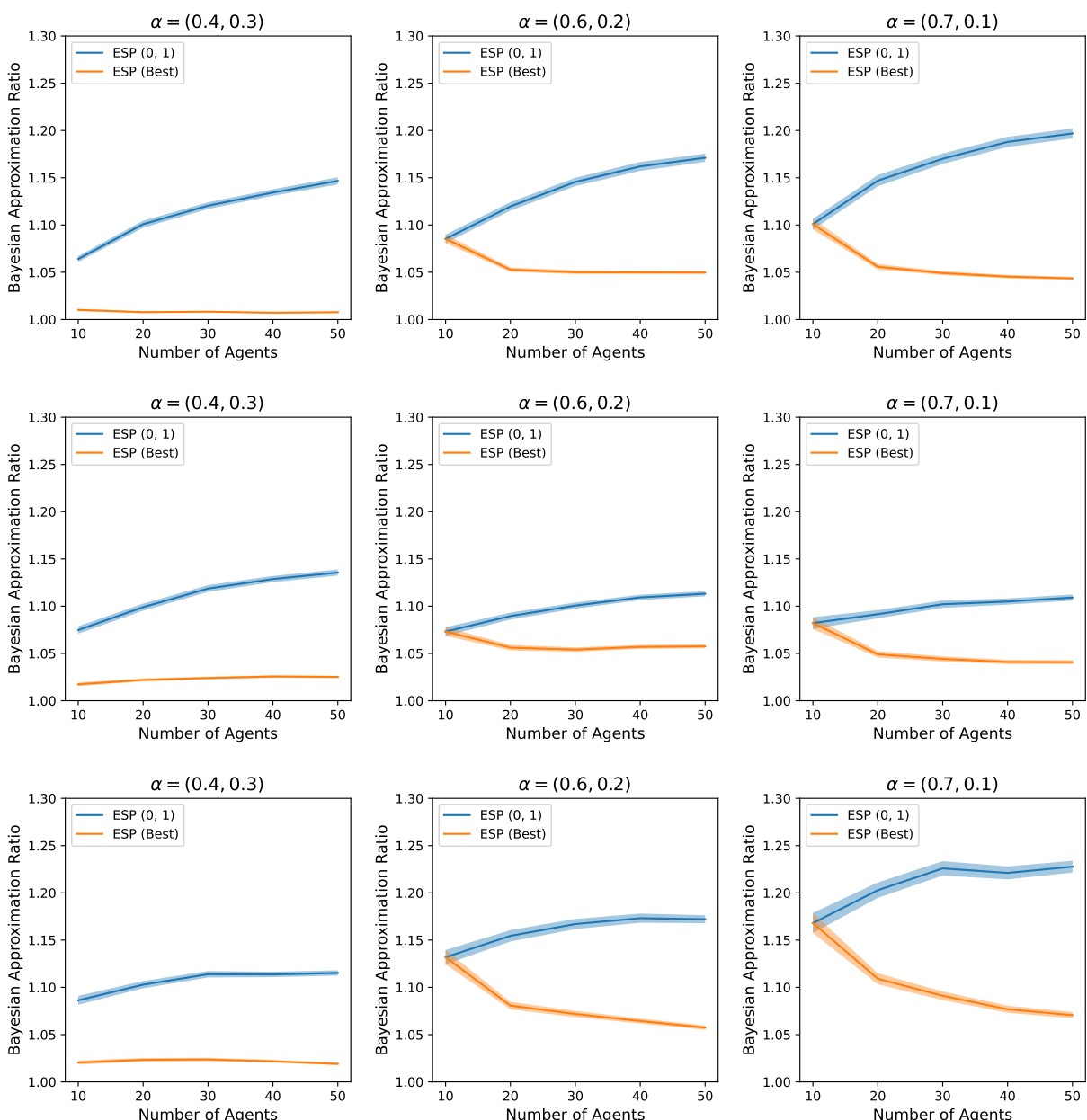

Figure 5: The Bayesian approximation ratio of $\mathcal{PM}_{best}$ and $\mathcal{PM}_{\vec{w}}$ when the agents are distributed according to $\mathcal{T}$ and the facilities are unbalanced, i.e. $k_1 = \alpha_1 n \neq k_2 = \alpha_2 n$ for $n = 10, 20, \ldots, 50$. Every column contains the results for different vector $\vec{k}$. The first row contains the results for a symmetric Beta distribution, that is $\mathcal{B}(5,5)$. The second row contains the results for the triangular distribution $\mathcal{T}$. The last row contains the results for the Uniform distribution $\mathcal{U}$.

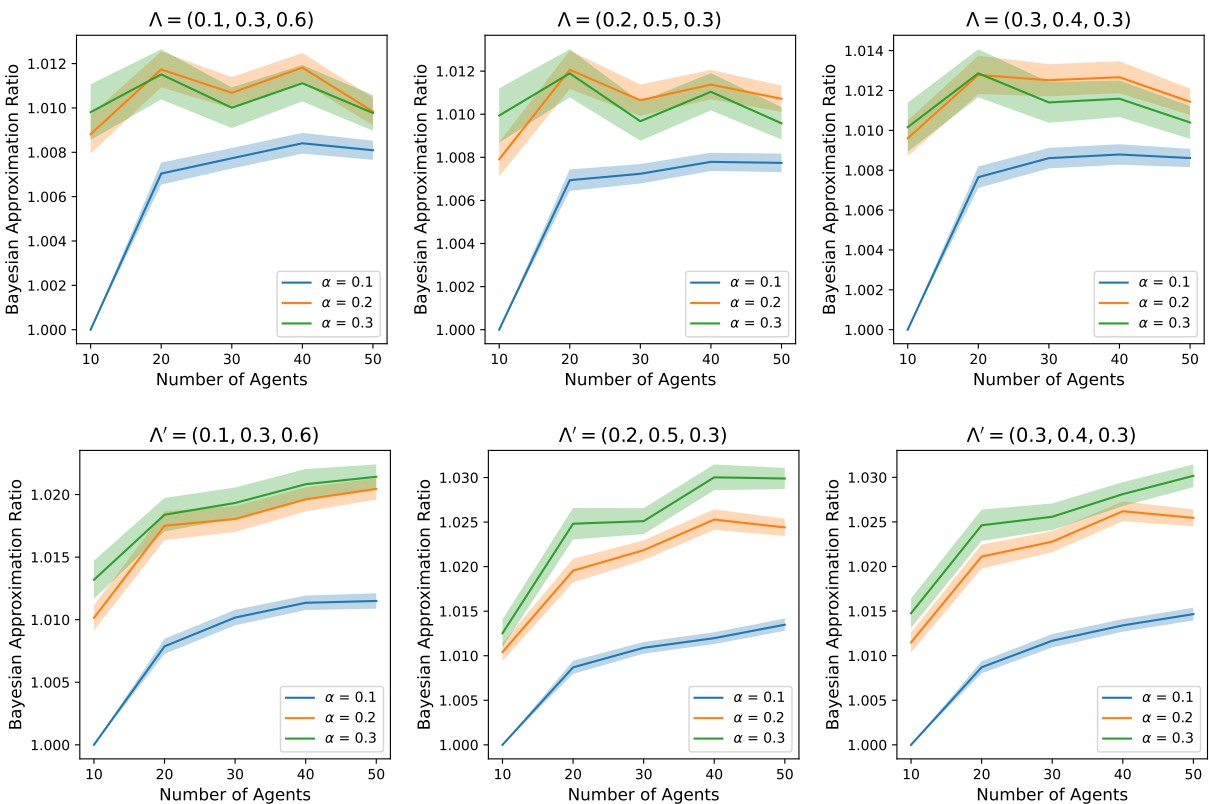

Figure 6: The Bayesian approximation ratio of $\mathcal{PM}_{best}$ for a population non i.d.. The capacities of the facilities are balanced, i.e. $k_1 = k_2 = \alpha n$ with $\alpha = 0.1, 0.2, 0.3$, and for $n = 10, 20, \ldots, 50$. In the first raw, the Beta distribution is symmetric, in particular $\mathcal{B}(5, 5)$, in the second raw the Beta distribution is asymmetric, in particular $\mathcal{B}(1, 9)$. Every column contains the results for different $\Lambda$.