# OpenReview forum: "On the Capacitated Facility Location Problem with Scarce Resources"
_auai.org/UAI/2024/Conference — UAI 2024 oral_

### Official Review · Reviewer_nsYA · 2024-03-12

**Q2-1 Originality-Novelty:** 2
**Q2-2 Correctness-Technical Quality:** 1
**Q2-5 Clarity Of Writing:** 3

**Q1 Summary And Contributions:**

The paper studies the capacitated facility problem with multiple facilities from a social welfare perspective. In more detail, in the capacitated facility location problem with m facilities (m-CFLP), n voters report an optimal point in the interval [0,1] and we are tasked to place m facilities with capacity constraints to serve them. In particular, the paper assumes that the facilities cannot serve all agents, which means that the assignment of agents to facilities also turns into a game (called the first-come-first-serve game) that needs to be taken into account when studying this scenario. Now, in this scenario, the authors focus on percentile mechanisms (which order the reports of the voters in increasing order and place each facility at a specific percentile) and first show that these mechanisms are absolutely truthful (which roughly means that they are even truthful when taking the FCFS game into account). Next, the authors restrict attention to the equilibrium stable percentile mechanisms (where each equilibrium of the FCFS game has the same social welfare) and study how well these mechanisms can approximate the optimal social welfare. To this end, the authors consider several cases analytically and also perform computer experiments.

**Q2-3 Extent To Which Claims Are Supported By Evidence:**

3: Good: the main claims are supported by convincing evidence (in the form of adequate experimental evaluation, proofs, (pseudo-)code, references, assumptions).

**Q2-4 Reproducibility:**

4: Excellent: key resources (e.g. proofs, code, data) are available and key details (e.g. proof sketches, experimental setup) are comprehensively described for competent researchers to confidently and easily reproduce the main results.

**Q3 Main Strengths:**

The problem considered in the paper (m-CFLP) certainly deserves attention as it brings facility location closer to practical applications. In particular, the combination of facility location with the FCFS game seems interesting (though I would appreciate more discussion of this design choice). The paper is also relatively well-written (except for quite a number of typos.) Lastly, I checked the proofs in the main body and found all of them (except for the one of Theorem 2) sound.

**Q4 Main Weakness:**

The paper has a number of significant weaknesses:

1) I find that some of the assumptions of the paper need to be discussed in more detail. Firstly, the authors consider utilities of the form $1-|x_i - y_i|$ rather than the typical cost of $|x_i-y_i|$, which makes quite a big difference in the approximation results. In particular, this means that the claims that adding capacity constraints results in bounded approximation ratios may be wrong (as the twist really is to maximize social welfare instead of minimizing social cost). Secondly, the authors never comment on why we need the FCFS game - in principle, our mechanism could also assign each agent to a facility and it is not apparent why this is a worse approach. Finally, I see little motivation (except of technical simplicity) to restrict attention to equilibrium-stable outcomes; why should this be a desirable property? In particular, if mechanism 1 is equilibrium-stable and mechanism 2 is not, but every equilibrium of mechanism 2 has higher social welfare than those of mechanism 1, I still would prefer mechanism 2.

2) The proof of Theorem 2 seems broken to me, as the effects of strategic behavior also need to be taken into account for the FCFS game. In particular, consider the following example with two facilities with a capacity of 2 each. We consider the percentile mechanism defined by v=(0, 0.5) and have 5 voters reporting positions (0.1, 0.2, 0.3, 0.4, 0.5). It is straightforward that our percentile mechanism will put the first facility at 0.1 and the second one at 0.3. Now, suppose that the voters report the following strategies for the FCFS game: (1,1,2,2,2). In particular, regardless of the strategy reported by voter 5, he will not be assigned to any facility, giving him a utility of 0. By contrast, assume that the last voter also reports 0.3 instead of 0.5. The outcome of our percentile mechanism will not change, but the FCFS game will now prefer him to the agent reporting 0.4. Hence, he can improve his utility and absolute truthfulness fails. This somewhat ruins the storyline of the paper because, without truthfulness, we could consider the mechanisms that optimize social welfare directly.

3) As a last critique point, I want to mention that I find the computer experiments in the fourth Section not particularly exciting. In particular, even without experiments, I find it not surprising that the (0,1) percentile mechanism does not perform particularly well with respect to social welfare (compared to most other mechanisms, not only the one the authors deemed as best). Moreover, also the results in Section 3.2.2 seem not particularly interesting to me as the approximation ratio of the given algorithms does not decrease even if we increase the number of facilities.

**Q5 Detailed Comments To The Authors:**

Detailed:
I want to mention that there is a rather high number of stylistic/ language issues in the paper (subsequently, I list some, not all):
-abstract: "When the number of facilities, m>1" -> When the number of facilities is m>1
-abstract (and elsewhere): there should be a comma before and after "i.e."
-During the first two pages, the objective for the approximation ratio is never explained. Similarly, the FCFS game is not explained. Such explanations are necessary to understand the introduction.
-page 2: "Social welfare" -> lower case
-page 2: "placement have different value" -> values
-page 2: "Game Theoretic" -> lower case
-page 2: "Following this initial study, in (Walsh [2022]) the authors..." -> Following this initial study, Walsh [2022] ...
-page 3: "that j-th facility can" -> that the j-th facility ...
-page 3: "prefixed priority rule" -> this is never mentioned before or after. In particular, explain what you mean by this.
-page 3: "the social welfare of the game is well defined" -> the social welfare is only defined for equilibria, I guess the authors mean here that the social welfare of all equilibria of the game is the same.
-Example 1: The computed social welfares seems wrong (the second and fourth agents already give a social welfare of 2...).
-Definition 2.1: I'd appreciate some intuitive explanation of what this definition says.
-Page 5: "the other cases follows" -> the other cases follow
-Page 5: the assumption that $k_1\geq k_2$ is never mentioned before
-Page 5, last sentence on the left column: break the sentence in multiple parts as it is too long to parse
-Can the authors explain the rather different behavior for \alpha=0.3n to \alpha=0.2n in Figure 3?

**Q9 Complying With Reviewing Instructions:**

Yes

---

> ### Author Rebuttal · Authors · 2024-04-04
>
> First, we want to thank the reviewer for the care and attention spent in assessing our paper. Should our paper be accepted, we pledge to correct all the typos that have been pointed out in the review.
>
> We reply to the questions down below.
>
> ---
>
> Q: I find that some of the assumptions of the paper need to be discussed in more detail. Firstly, the authors consider utilities of the form 1-|x_i-y_j| rather than the typical cost of |x_i-y_j|, which makes quite a big difference in the approximation results.
>
> A: On the one hand, we want to stress that we addressed an open question raised in a previously published paper, namely Aziz et al.'s work on "The Capacity Constrained Facility Location Problem". The reason they chose to consider the utility rather than the cost of the agents is that it is not clear what the cost of an unserved agent should be.
>
> On the other hand, it is worth noting that evaluating the performance of a mechanism based on the total utility of the agents is standard practice. We refer the reviewer to the following papers for further references:
>
>    - "Facility Location Games with Dual Preference" by S. Zou and M. Li;
>    - "Approximately Optimal Mechanism Design via Differential Privacy" by K. Nissim, R. Smorodinsky, and M. Tennenholtz
>    - "Deep Learning for Multi-Facility Location Mechanism Design" by N. Golowich, H. Narasimhan, and D.C. Parkes.
>
> ---
>
> Q:  Finally, I see little motivation (except of technical simplicity) to restrict attention to equilibrium-stable outcomes; why should this be a desirable property?
>
> A: Having a mechanism whose outputs are equilibrium-stable is appealing because it enables us to define the social welfare of the mechanism for every input. From a practical viewpoint, this is useful as it allows the mechanism designer to forecast the social welfare generated by the mechanism and predict agents' behavior. Notice that, even if this property might seem restrictive, it enables us to define routines that are almost optimal.
>
> ----
>
> Q: The proof of Theorem 2 seems broken to me, as the effects of strategic behavior also need to be taken into account for the FCFS game. In particular, consider the following example with two facilities with a capacity of 2 each. We consider the percentile mechanism defined by v=(0, 0.5) and have 5 voters reporting positions (0.1, 0.2, 0.3, 0.4, 0.5). It is straightforward that our percentile mechanism will put the first facility at 0.1 and the second one at 0.3. Now, suppose that the voters report the following strategies for the FCFS game: (1,1,2,2,2). In particular, regardless of the strategy reported by voter 5, he will not be assigned to any facility, giving him a utility of 0. By contrast, assume that the last voter also reports 0.3 instead of 0.5. The outcome of our percentile mechanism will not change, but the FCFS game will now prefer him to the agent reporting 0.4. Hence, he can improve his utility and absolute truthfulness fails.
>
> A: The reviewer's example is incorrect. If the agent reports 0.3 instead of 0.5, it will not change the outcome of the First-Come First-Served (FCFS) game. The set of agents that will be served after the FCFS game depends on the strategy they choose, i.e., which $j$ they select from the set $[m]$, and their true type, not the type they reported. In particular, even if the agent reports 0.3 to the mechanism and plays 2 during the FCFS game, it will not affect the fact that two other agents are closer to the facility, hence the manipulative agent's utility will remain equal to 0.

---

### Official Review · Reviewer_77Aj · 2024-03-21

**Q2-1 Originality-Novelty:** 2
**Q2-2 Correctness-Technical Quality:** 3
**Q2-5 Clarity Of Writing:** 3

**Q1 Summary And Contributions:**

The paper deals with an open problem proposed by Aziz et al (Games and Economic Behavior, 2020) related to a strategic version of the facility location problem.  In this problem we have to locate m resources on a line, where each each resource has a limited capacity to serve requests and the total number of users is assumed to be greater than the total capacity of the resources (and thus some users will  be unsupported). Moreover, given the locations of the resources each user can choose the resource to access and each resource serves the closest users limited from its capacity. Thus, users are engaged in a game to choose the resource to access.
it is known that when there is only one resource this game has only on Nash Equilibrium. When there are more resources the game played by the users for selecting the resources has several Nash Equilibria and the utility of each player depends on the NE of this game. This means that the classic notion of truthfulness is not available for this game.

The main contributions of the paper are the introduction of a stronger notion of truthfulness, called absolute thruthfullness, and the definition of a class of mechanisms, called Equilibrium Stable mechanisms, where all the NE of the game achieve the same social welfare. Then, it is shown that percentile mechanisms are absolutely truthful and, under some conditions, they are also Equilibrium Stable. Moreover,  an analysis is presented to select the percentile mechanism that has the best approximation factor among the ES mechanisms.

Finally, some experiments are presented to empirically study the behavior of the best percentile mechanisms.

**Q2-3 Extent To Which Claims Are Supported By Evidence:**

3: Good: the main claims are supported by convincing evidence (in the form of adequate experimental evaluation, proofs, (pseudo-)code, references, assumptions).

**Q2-4 Reproducibility:**

3: Good: key resources (e.g. proofs, code, data) are available and key details (e.g. proofs, experimental setup) are sufficiently well-described for competent researchers to confidently reproduce the main results.

**Q3 Main Strengths:**

The paper deals with an interesting problem and it solves an open problem proposed by Aziz et al. (Games and Economic Behaviors, 20209.

**Q4 Main Weakness:**

The organization of the paper and the clarity of the presentation should be improved.

**Q5 Detailed Comments To The Authors:**

Why is the game called FCFS even if the served players are the closest ones?

Which is the relation between your definition of absolutely truthful mechanisms and dominant strategy?

What is a percentile vector? Maybe it could be helpful to add an example of these mechanisms.

Notation in Definition 2.2 is inconsistent with the rest of the paper.

I guess, there is an error in Example 1 and the SW should be 3.6 and 3.5.

**Q9 Complying With Reviewing Instructions:**

Yes

---

> ### Author Rebuttal · Authors · 2024-04-04
>
> First, we want to thank the reviewer for the care and attention spent in assessing our paper. Should our paper be accepted, we pledge to correct all the typos that have been pointed out in the review.
>
> We reply to the questions down below.
>
> ---
>
> Q: Why is the game called FCFS even if the served players are the closest ones?
>
> A: The name was introduced by Aziz et al. in "The Capacity Constrained Facility Location Problem".
>
> The name comes from the fact that if a facility with capacity $k$ opens, it will serve the first $k$ agents that reach the facility, leaving the others unserved. Since we have a single facility to place and the agents are assumed to be homogeneous (i.e., they have the same speed), the first $k$ agents reaching the facility do coincide with the $k$ agents that are closer to the facility.
>
> However, notice that when there is more than one facility to locate, the set of agents served by the facilities does not necessarily coincide with the set of agents that are closer to the facilities. This can be seen in Example 1 in our paper, where two different equilibria have different sets of agents served (hence the set of agents served is not fixed).
>
> ---
>
> Q: Which is the relation between your definition of absolutely truthful mechanisms and dominant strategy?
>
> A: A mechanism is absolutely truthful if reporting the real position is a dominant strategy for every agent, regardless of what the other agents' real positions is and, more crucially, regardless of what the strategies the other agents employ in the First-Come First-Served (FCFS) game are.
>
> Notice that using this novel concept of truthfulness is necessary because the equilibrium of the FCFS game is in general not unique and, consequentially, we need to fix the strategies of the other agents in order to compare the different payoffs of the agent in a meaningful way.
>
> ---
>
> Q: What is a percentile vector? Maybe it could be helpful to add an example of these mechanisms.
>
> A: Percentile vectors are vectors containing parameters that induce the percentile mechanisms, as described in the paper "Analysis and Optimization of Multi-dimensional Percentile Mechanisms" by Sui et al.. If the paper is accepted, we will add an example to better illustrate the mechanism routine for those who are not familiar with it.

---

### Official Review · Reviewer_kB1J · 2024-03-21

**Q2-1 Originality-Novelty:** 3
**Q2-2 Correctness-Technical Quality:** 3
**Q2-5 Clarity Of Writing:** 2

**Q1 Summary And Contributions:**

This paper addresses the m-capacitated facility location problem where the total facility capacity is lower than the number of agents. Specifically, after locating m facilities, agents compete for access to these facilities through a first-come-first-serve game. The objective is to devise a truthful mechanism with a high approximation ratio compared to the optimal social welfare.
To address the possibility of multiple Nash equilibria in the first-come-first-serve game, the authors introduce two key concepts: absolutely truthfulness and equilibrium stability (ES). They demonstrate the existence of an absolutely truthful and ES percentile mechanism for m=2 and characterize its approximation ratio. For m>2, they establish an asymptotically tight approximation ratio when the number of agents n is sufficiently large. Additionally, the authors provide empirical evaluations of their results.

**Q2-3 Extent To Which Claims Are Supported By Evidence:**

4: Excellent: all claims are supported by very convincing evidence (in the form of comprehensive experimental evaluation, rigorous mathematical proofs, detailed (pseudo-)code, precise references, well-motivated and realistic assumptions) and the authors deliver what they promise.

**Q2-4 Reproducibility:**

4: Excellent: key resources (e.g. proofs, code, data) are available and key details (e.g. proof sketches, experimental setup) are comprehensively described for competent researchers to confidently and easily reproduce the main results.

**Q3 Main Strengths:**

+ This work addresses a natural generalization of Aziz et al. (2020b) to the case where m>=2, which is an important extension. The problem tackled here is significant and deserves attention.
+ The overall structure of the paper is well-organized, and the proof of the approximation ratio appears to be non-trivial.

**Q4 Main Weakness:**

- This paper primarily focuses on the case where the total capacity is less than the number of agents. While this assumption is natural when dealing with a single facility, for the multiple facilities case, I believe the case where the total capacity exceeds n is also significant and merit consideration.

**Q5 Detailed Comments To The Authors:**

1. Could you provide justification for assuming that the total capacity is less than the number of agents? Alternatively, is there a reduction demonstrating that the case where the total capacity exceeds n is simpler?
2. Your work predominantly relies on the percentile mechanism for all results. While this choice aligns naturally with optimizing social welfare only, I am curious whether there are alternative mechanisms offering better approximation guarantees while also ensuring truthfulness.

Some minor comments:
1. in the last paragraph in Page 1, I think the focused framework should be in Aziz et al. (2020b) but not (2020a).
2. in the last paragraph in Page 2, I think some descriptions of the difference between your work and Aziz et al. (2020b) are missing.
3. in example 1, the number should be ‘3.6 3.5’ but not ‘1.6 1.5’.
4. in the proof of Thm 3, the definition of B_{r_j}(y_j) is missing.
5. in the proof of Thm 7, ‘i_1 i_m>=…’, a typo.
6. in Mechanism 1, ‘a+2mk<=b<=n’, a typo. I think this should be ‘a+mk’.

**Q9 Complying With Reviewing Instructions:**

Yes

---

> ### Author Rebuttal · Authors · 2024-04-04
>
> First, we want to thank the reviewer for the care and attention spent in assessing our paper. Should our paper be accepted, we pledge to correct all the typos that have been pointed out in the review.
>
> We reply to the questions down below.
>
> ---
>
> Q: Could you provide justification for assuming that the total capacity is less than the number of agents? Alternatively, is there a reduction demonstrating that the case where the total capacity exceeds n is simpler?
>
> A: The idea of modelling agents that compete for a scarce resource through a First-Come, First-Served (FCFS) game was introduced by Aziz et al. in "The Capacity Constrained Facility Location Problem." When the number of agents is smaller than the total capacity of the facilities, the purpose of the FCFS game falls apart. This is demonstrated by the fact that the case in which we have enough capacity to serve all the agents has been approached differently (i.e. the mechanism assigns the agents to the facilities, see "Facility location problem with capacity constraints: Algorithmic and mechanism design perspectives" by Aziz et al. or "Strategy proof mechanisms for facility location at limited locations" by Walsh). In this case there is no FCFS game because the agent-to-facility-assignment is decided by the mechanism.
>
> One could try to study what happens if the agent-to-facility is not fixed by the mechanism but is decided via the FCFS game as it happens in our paper. However, in this case the percentile mechanisms either place the facilities at the same spot or they are not absolutely truthful.
>
> ---
>
> Q: Your work predominantly relies on the percentile mechanism for all results. While this choice aligns naturally with optimizing social welfare only, I am curious whether there are alternative mechanisms offering better approximation guarantees while also ensuring truthfulness.
>
> A: The reviewer is completely right to point this out: it would be interesting to investigate whether it would be possible to improve the approximation ratio results by considering different routines. However, notice that this study should be more oriented toward cases in which we need to place two facilities, as in these instances the gap between our approximation ratio guarantees and 1 is larger, and percentile mechanisms are quasi-optimal when $m$ becomes large.

---

### Official Review · Reviewer_X7aC · 2024-03-22

**Q2-1 Originality-Novelty:** 3
**Q2-2 Correctness-Technical Quality:** 3
**Q2-5 Clarity Of Writing:** 4

**Q1 Summary And Contributions:**

In this paper, the authors study mechanism design aspects of the m-Capacitated Facility Location Problem (on a single dimension). They study the case where the total capacity is less than the number of agents. In this problem, the utilities and social welfare are determined via a First-Come-First-Served game. They study two properties of mechanisms: absolute truthfulness, and equilibrium stability. They identify a class of mechanisms (percentile mechanisms) and show that these are absolutely truthful. Moreover, they pinpoint the conditions under which they are equilibrium stable. They also study the approximation ratio (the worst-case ratio between the objective achieved and the optimum). Finally, they provide experimental results that investigate the performance of the percentile mechanism from a Bayesian perspective.

**Q2-3 Extent To Which Claims Are Supported By Evidence:**

4: Excellent: all claims are supported by very convincing evidence (in the form of comprehensive experimental evaluation, rigorous mathematical proofs, detailed (pseudo-)code, precise references, well-motivated and realistic assumptions) and the authors deliver what they promise.

**Q2-4 Reproducibility:**

4: Excellent: key resources (e.g. proofs, code, data) are available and key details (e.g. proof sketches, experimental setup) are comprehensively described for competent researchers to confidently and easily reproduce the main results.

**Q3 Main Strengths:**

- The authors study a natural and relevant problem
- The authors provide a solid mechanism design analysis
- The results are described clearly

**Q4 Main Weakness:**

(none)

**Q5 Detailed Comments To The Authors:**

The paper is well written. The problem that the authors set out to solve is described clearly, and connections to related work are described well. The technical results follow a clear and natural story line. The authors motivate why the different technical definitions make sense. The results are interesting and provide a useful mechanism design view on the setting. Overall, I find that the paper provides a solid contribution.

**Q9 Complying With Reviewing Instructions:**

Yes

---

> ### Author Rebuttal · Authors · 2024-04-05
>
> We thank the reviewer for the care and effort put into assessing our paper.

---

### Official Review · Reviewer_L1eh · 2024-03-22

**Q2-1 Originality-Novelty:** 3
**Q2-2 Correctness-Technical Quality:** 4
**Q2-5 Clarity Of Writing:** 4

**Q1 Summary And Contributions:**

The paper considers social choice theory and addresses the facility location problem FLP where agents are assigned
to facility in first-come first serve game after the facilities have been located, and Nash equilibria for the social welfare
are computed. The additional complexity is that the facilities are capacitated, so that they cannot host all the agents.
The paper comes with several theoretical analyses of the m-CFLP problem and an experimental work based
on Bayesian reasoning. Limits of the approximation ratios are studied.

**Q2-3 Extent To Which Claims Are Supported By Evidence:**

4: Excellent: all claims are supported by very convincing evidence (in the form of comprehensive experimental evaluation, rigorous mathematical proofs, detailed (pseudo-)code, precise references, well-motivated and realistic assumptions) and the authors deliver what they promise.

**Q2-4 Reproducibility:**

3: Good: key resources (e.g. proofs, code, data) are available and key details (e.g. proofs, experimental setup) are sufficiently well-described for competent researchers to confidently reproduce the main results.

**Q3 Main Strengths:**

The writing is almost flawless and the theory is strong.

The theory includes results that at least on pure NE exists (no mixed strategies), and a direct strategy for two facilities.
refering to the percentile mechanism PM from Sui et al. [2013] they show that all PMs are absolutely truthful..
They also analyze equilibrium stable (ES) PMs and give an approximation ratio for two and more facilities.

Facility location has a long tradition and is, e.g., used as one component in "location-routing" as in

Y. Warsame, "Integrating Location-Routing with Task and Motion Planning," 2020 IEEE 16th International Conference on Automation Science and Engineering (CASE), Hong Kong, China, 2020, pp. 329-334, doi: 10.1109/CASE48305.2020.9216819.

**Q4 Main Weakness:**

Its not no uncertainty in the m-CLFP and in the theory, so the relevance to UAI is only in applying Bayesian methods in the experiments.
There is a percentile mechanism, but as far as I can see it is not uncertain.

The problem is very specialized for the general audience to appreciate.

Even for me as a reviewer all the notation of ES, absolutie truthfulness, percentile mechanisms, social welfare are difficult to grasp in their entirety and interplay.

The paper is not self-contained, as they rely on previous work of Aziz et al. (2020) that connects facility location problem to social welfare.

There is an 2024 AAMAS paper "Extended ranking mechanisms for the m-capacitated facility
location problem in bayesian mechanism design" (maybe by the same authors), where the review had no access to check for overlap

The author could make at least one figure to introduce the problem, and cite who else was using it.

**Q5 Detailed Comments To The Authors:**

[ Combination of the above ]

**Q9 Complying With Reviewing Instructions:**

Yes

---

> ### Author Rebuttal · Authors · 2024-04-05
>
> We thank the reviewer for the care and effort put into assessing our paper.
>
> If the paper gets accepted, we pledge to implement the suggestions detailed by the reviewer.

---

### Meta-Review · Area_Chair_tD1c · 2024-04-15

5 high-quality reviews are obtained. One reviewer recommends strong acceptance, three reviewers recommend acceptance, and one reviewer recommends weak acceptance. Overall, all reviewers are very positive about the paper, with a good level of confidence in reviewing the paper. The paper is well-written, technically solid, and contributes well to the UAI.